# The perceived barriers and facilitators for chiropractic care in older adults with low back pain; insights from a qualitative exploration in a dutch context

Lobke P. De la Ruelle[1]*, Annemarie de Zoete[1], Cornelius Myburgh[2,3,4], Hella E. Brandt[1], Sidney M. Rubinstein[1]

1 Department of Health Sciences, Faculty of Science, Amsterdam Movement Sciences, Vrije Universiteit Amsterdam, Amsterdam, The Netherlands, 2 Department of Sports Science and Clinical Biomechanics, University of Southern Denmark, Odense, Denmark, 3 The Chiropractic Knowledge Hub, University of Southern Denmark, Odense, Denmark, 4 Department of Chiropractic, University of Johannesburg, Johannesburg, South Africa

* l.p.delaruelle@vu.nl

**Data Availability Statement:** All relevant data are within the paper and its Supporting information files. The transcripts of the interviews and focus

## Abstract

### Background

Understanding care seeking behaviour is vital to enabling access to care. In the context of low back pain (LBP), chiropractors offer services to patients of all ages. Currently, geriatric sub-populations tend to be under-investigated, despite the disproportionate effects of LBP on older adults. In the Netherlands, the chiropractic profession is relatively unknown and therefore, generally speaking, is not considered as the first choice for conservative musculo-skeletal primary health care. The aim of this paper was to explore the experiences of older adults with LBP, seeking chiropracic care for the first time, in order to identify perceived barriers and facilitators in this process.

### Methods

Stage 1: Participants 56 years of age and older with chronic LBP who either sought or did not seek chiropractic care were interviewed to provide detailed information on the factors that promoted or impeded care-seeking behaviour. A purposive sampling strategy was used to recruit participants through a network of researchers, chiropractors and other healthcare professionals offering musculoskeletal health care services. Individuals with underlying pathology, previous surgery for LBP, or insufficient mastery of the Dutch language were excluded. Data were collected until saturation was reached and thematically analysed. Stage 2: To further explore the themes, a focus group interview was conducted with a provider stakeholder group consisting of:two physiotherapists, a nurse practitioner, a geriatrician, and a chiropractor. All interviews were conducted online, voice recorded, and transcribed verbatim.

group cannot be shared publicly due to privacy reasons as participants can be identified. This is in accordance with European Data Protection Regulation (GDPR) (https://libguides.vu.nl/rdm/gdpr-privacy). Individual participants' data are available upon request from the principal investigator (Sidney Rubinstein, s.m.rubinstein@vu.nl) or the secretary of the department of Health Sciences (secretariaat.agw.beta@vu.nl) for researchers who meet the criteria for access to confidential data.

**Funding:** This study is funded by European Centre for Chiropractic Research Excellence (ECCRE), located in Odense, Denmark (grant number 01-20160NL/MvT) as well as funded by the Nederlandse Chiropractoren Associatie (NCA), located in Emmeloord. The funders had no role in study design, data collection and analysis, decision to publish, or preparation of the manuscript.

**Competing interests:** Four authors (LDLR, ADZ, CM and SMR) are chiropractors who work in clinical practice, but have no direct financial interests linked with this survey. This does not alter our adherence to PLOS ONE policies on sharing data and materials.

## Results

We interviewed 11 older adults with low back pain. During this process four themes emerged that captured their perception and experiences in either seeking or dismissing chiropractic care for their LBP; these being 'generic', 'financial', 'expectation', and 'the image of the chiropractor'. The focus group members largely confirmed the identified themes, highlighting a lack of awarenes and accessibility as key barriers to care. On the other hand, whe chiropractior as an alternative care provider, with a focus on manual interventions, was seen as a facilitator.

## Conclusions

The lack of knowledge about chiropractic care was found to be the most important barrier to seeking care. The most important facilitator was insufficient resolution of their symptoms following previous care, making patients look further for a solution for their problem. These barriers and facilitators seem not to differ greatly from barriers and facilitators found among younger patients with neck pain. Age and health condition may therefore be weak determinants of care. This new information may help us optimize accessibility for older adults to the chiropractor.

## Introduction

Low back pain (LBP) is a highly prevalent musculoskeletal complaint in western countries, which is associated with substantial costs [1]. Given the aging population in the Netherlands, these figures are only likely to increase in the coming years. In comparison to their younger counterparts, LBP may have a disproportional effects on older adults due to a decreased mobility resulting in limited social participation and a greater chance of isolation, and reduced perceived quality of life [2, 3]. However, data is currently severely lacking in this sub-population.

In the Netherlands, chiropractic care is considered 'alternative care'. For the Dutch LBP patients this means that health care insurance only covers some, but never 100% of the costs. The profession is relatively small with only 571 practitioners in a country with more than 17 million inhabitants. Despite their small numbers, in 2019 more than 1 million treatments were delivered by chiropractors (unpublished observations) in the Netherlands. All chiropractors working in the Netherlands are trained abroad, as there is no chiropractic college in the country.

While much is known about the effects of chiropractic treatment and low back pain (LBP), far less is known about what actually motivates individuals to seek care. Many LBP patients do not seek care from a health care practitioner for their complaints [4–6]. Most studies performed on care-seeking behaviour for LBP highlighted characteristics associated with the choice of care. The intensity of pain and disability, the fear of future job impairment, involvement in sports, age and income and education levels were determinants found to influence care seeking behaviour for LBP [4, 5]. However, besides these unmodifiable factors, patients also appear to be influenced in their choice of health care practitioners by additional factors. Older adults are, however, less likely to seek care for their LBP, due to the assumption that LBP is a normal part of ageing. Moreover, older individuals tend to display aa negative attitude towards pharmacological and/or surgical interventions and deem their LBP less important than other comorbidities [7, 8]. Presently, however, a paucity in the literature appears to exist

with respect to an exploration the barriers and facilitators perceived by the patients when seeking care from a chiropractor for a LBP-related problem.

Several qualitative studies have been conducted on patients' experience of spinal manipulative therapy (SMT) and chiropractic maintenance care [9–12]; however, none of these studies focused on the barriers and facilitators for chiropractic care in older adults with LBP. To our knowledge, only one study has examined the barriers and facilitators [11] for those with neck pain undergoing spinal manipulative therapy by a manual therapist; therefore, it is unclear how this relates to chiropractic care and LBP, particularly the elderly [11]. For example, older adults might have more fear of SMT, which in-turn may be less for treatment of the low back than the neck. Therefore, it is important to investigate barriers and facilitators in older adults with LBP seeking chiropractic care in the Netherlands.

The objective of this study was to identify factors, specifically the barriers and facilitators, pertinent in care-seeking behaviour among older LBP patients without prior experience with chiropractic care when seeking care from a chiropractor.

## Methods

### Design

We constructed an explorative, qualitative case study, with the case defined as old individuals engaged in the process of care seeking for a lower back pain-related problem [13]. This approach was deemed appropriate to develop a thick description of individual and shared experiences occurring during this process; both individuals who had chosen chiropractic care and those who did not. The 32-item checklist for interviews and focus groups [Consolidated criteria for reporting qualitative research (COREQ)] was used as a guideline for the report [14].

### Theoretical approach

From a theoretical perspective, we drew from a recent comprehensive literature review conducted by a Dutch research group [15]. In their investigation, Hupkens et al. highlight several processes focused on maintaining a high quality of life, living independently and being to engage and participate in society as integral to attaining meaning in life in older age. Our position, although perhaps tacit, was that LBP represents a factor that has the potential to rob older adults of their highly valued needs if their care seeking is unsuccessful.

Our definition of what age might define an older person was pragmatic in nature, and was aimed at alignining with current published literature from within the discipline of chiropractic. In this regard 55 years of age is considered 'older' [16].

Our study was also underpinned by a constructivist stance, as we sought to create a meaningful co-construction of pathways to careseeking through the experiences of participant stakeholders [13].

### Participants

**Inclusion criteria.**   Adults older than 55 years of age, suffering with LBP (with or without radiation to the legs), were invited to participate. As this study is part of a larger study (BACE-C), the age-related inclusion criterium was based on this study [16].

**Exclusion criteria.**   Those who previously visited a chiropractor were excluded. In addition to those with underlying pathology andprevious surgery of the low back were excluded to limit the variation in severity of the complaint. Also insufficient mastery of the Dutch language was an exclusion factor.

## Sampling

Purposive sampling was used, meaning we included participants with different genders, ages, and socioeconomic status as well as from different areas of the country [17]. To achieve this, potential participants were contacted if they balanced these factors out.

As we wanted to gain insight on the barriers and facilitators from a spread of elderly LBP sufferers, we contacted both participants with LBP, who had an appointment with a chiropractor, but had not yet been for their first consultation, and participants with LBP who had notsought chiropractic care.

Chiropractors in good-standing with the NCA (Dutch Chiropractic Association) were asked to recruit potential participants. The potential participants that did not seek chiropractic care were contacted by the research team or their network.

Potential participants were contacted by telephone by the interviewer to make sure they fulfilled the inclusion criteria and to schedule a video call using Zoom [18]. Information was sent to the participants by e-mail and written and verbal informed consent was obtained. The participants were given the opportunity to ask questions at the beginning and the end of the interview and were assured that they could withdraw from the interview at any time. Apart from the e-mail and telephone contact, the participants and interviewer had never been in contact previously. The participants were only interviewed once.

## Data collection

The study was conducted in two stages. Stage one consisted of semi-structured interviews with LBP sufferers during the period of February 2021 to September 2021. As a continuation of thematic exploration, data were seubsequentlycollected through a focus group interview during May 2022.

**Stage 1: Interviews of LBP patients over 55 years of age.** Individual interviews were conducted by a female chiropractor and early-career research (LPR), and explored the process of care seeking, with a focus on factors that tend to frustrate or facilitate the care seeking process.

The interviewer followed a course 'qualitative research in the field of health care' with six hours of lectures and nine hours of practical workgroups. After four initial interviews, transcripts were evaluated by the project group and adaptations made to the interview guide. Four questions were added to aid the participants in answering questions with greater reflective depth, whilst still keeping the interviews well bracketed. Technical elements of the interview process were also discussed in order to assist the researcher in developing her interviewing style. All interviews were conducted in Dutch via online video call (Zoom), were voice recorded, and transcribed verbatim. The transcriptions were generated by the investigators, chiropractic colleagues or by a transcription company. Quotes used in the present article were translated contextually into English by a person fluent in both languages. Using an iterative approach, topic guides (S1 Appendix) were initially used and subsequently revised to distill emergentthemes [17].

Data were collected until code saturation occurred and no new themes emerged.

**Stage 2: A focus group with relevant health care professionals.** In this second stage, a focus group was conducted consisting of five relevant stakeholders, these being: a physiotherapist researcher, a geriatric physiotherapist, a chiropractor, a nurse practitioner, and a geriatrician. All individuals were purposively recruited via the network of the research team. A topic guide, based on the individual interviews was usedto help further unpack the themes identified in the first stage, thus helping to generate a further, more nuanced understanding of their meaning. No patients were invited to participate in the focus group to obtain the themes from an 'outsider' perspective. The group members were initially asked what they

thought the older adults would perceive as barriers and facilitators for seeking care from a chiropractor for their LBP. Following the groups feedback, their answers were compared with the participant feedback gleened from the individual interviews. The focus group was held online in Dutch, voice recorded, and transcribed verbatim. The duration of the focus group was 90 minutes.

Our strategy in phase 1 and 2 was aimed at optimizing data completeness [19]. Specifically, interviews allowed for a thick description of individual experiences, whilst focus groups captured a range of barriers and facilitators experienced in the responder group which accentuated differences and similarities. This methodological strategy is thought to optimize qualitative-qualitative triangulation, which comprises the combination of different qualitative data collection methods.

## Data analysis

The first 4 interviews with participants seeking care and the first 2 interviews with participants not seeking care were independently coded by two investigators (LPR and ADZ) using open coding and later discussed between them until mutual agreement to increase intercoder agreement. All other interviews were coded solely by the main investigator (LPR). The codes were analyzed and situated in a thematic map in MaxQDA 2020 [20].

A summary of each transcript was made based on the first analyses =. These were subsequently made available to the relevant participants to ensure their accuracy (member check) [17]. Nine participants returned member check with full agreement.

Data were organized as follows: (1) characteristics of participants and their LBP; (2)Treatment history (which reflected an overview of the care-seeking behaviour); and (3) Barriers and facilitators of seeking chiropractic care.

A thematic analysis was performed. This process involved codifying and then abstracting connected data to common themes. Initially, line-by-line inductive coding was performed to define and develop a codebook. Two study collaborators acted as code checkers, providing critical feedback and subsequent refinement of the code book. The code book was then used to code subsequent interviews deductively. The emerging themes were discussed in meetings with the entire research group until the barriers and facilitators were identified when a consensus was reached.

## Ethical considerations

The medical ethical committee of VU University Medical Center Amsterdam approved the study and declared that the study does not fall within the scope of the Medical Research Involving Human Subject act (ref. number 2017.618).

## Results

Sixteen individuals were contacted to participate, of which 4 were excluded (having had surgery of the back (n = 2) or having seen a chiropractor in the past (n = 2)) and one declining to participate because she did not have access to a computer. Of these 5 people, 3 were female and 2 male. All interviews only had the interviewer and participant present except for one, where the spouse was also present in the background.

In total, we interviewed 11 participants, of which 8 had made an initial appointment with a chiropractor and 3 did not seek chiropractic care. Six participants were male, 5 participants were female and the ages ranged from 57 to 90 years of age (see also Table 1). The interviews lasted on average 62 minutes (ranging from 33 to 101 minutes).

**Table 1. Patient characteristics.**

| Participant | Age | gender | Duration LBP | How limiting | Underlying conditions | Imaging | Education level | Profession |
|---|---|---|---|---|---|---|---|---|
| 1 | 76 | M | 15y | Limited in walking duration | Myasthenia gravis | X-ray | University | Project leader/ Engineer |
| 2 | 81 | F | 15-20y | Limited in ADL, cannot walk without aid | Uterine problems, breast amputation, cardiovascular problems | X-ray | IVE | Housewife |
| 3 | 69 | M | 3y | Not limited, struggles with some movements | Had one epileptic seizure | - | University | Social Psychologist |
| 4 | 59 | M | 40+y | Limited in heavier tasks, like gardening | Reumathoid arthritis | X-ray | HVE | Project leader |
| 5 | 66 | M | 5-6y | Limited in sports and heavy lifting | - | - | HVE | Teacher/ headmaster primary school |
| 6 | 77 | F | 30y | Limited in walking duration | Degenerative joint disease | - | HVE | Nurse |
| 7 | 90 | M | 30y | Severely limited in walking -duration | Balance problems | Dexa-Scan | IVE | Office Clerk |
| 8 | 58 | F | 15-20y | Not limited, more careful | Intestinal problems | - | HVE | Sales Representative |
| 9 | 60 | F | 20-25y | Limited in heavy lifting | - | X-ray | IVE | Housewife |
| 10 | 57 | F | 10y | Not limited, adapts heavier tasks | - | MRI | Secondary Education | Administrative worker |
| 11 | 64 | M | 2y | Limited in heavy lifting | Heart rhythm disorder | MRI | HVE | Financial controller |

## Description of pathways of care-seeking behaviour

The participants had sought various different forms of care for their LBP. Fig 1 demonstrates the different pathways to healthcare providers the participants had followed up to the moment of the interview. Most participants had previously contacted (1) other therapists (e.g. physiotherapists, acupuncturists, osteopaths, manual therapists) directly or with a referral from their GP, (2) their GP (primarily prescribing medication, referring or ordering imaging), or (3) medical specialist, referred to by the GP. For those that sought chiropractic care, one participant was referred to the chiropractor by their GP, all other participants contacted the chiropractor directly.

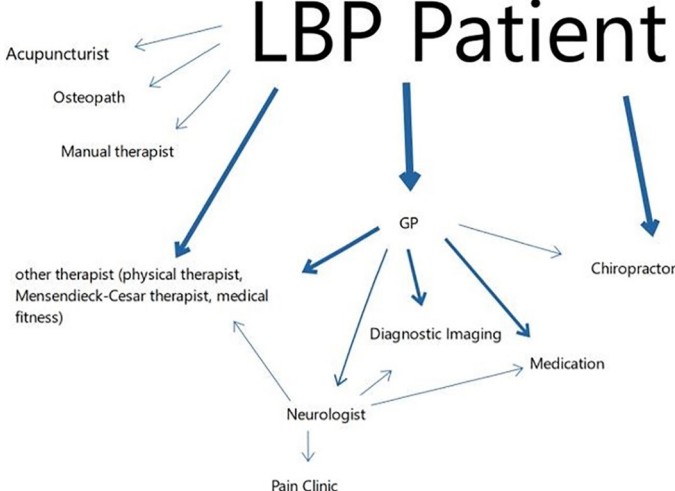

**Fig 1. The pathways the participants of this study took up to the interviews.** The thickness of the arrows indicates the proportion of participants that followed this pathway.

The participants sought chiropractic care via different pathways (e.g. walked passed a clinic, word of mouth, media, or their GP). When seeking information on chiropractic care, the participants resorted to their GP, the internet, leaflets, their insurance company, or acquaintances.

## Thematic results: Barriers and facilitators

Four factors were found to influence the care-seeking behaviour of older adults with LBP (Fig 2). These factors are organized into 4 themes: 1) 'generic aspects', 2) 'financial', 3) 'expectations', and 4) 'image chiropractor'. The barriers and facilitators are found in every theme and most subthemes were mentioned both as a barrier and as a facilitators, such as: the awareness, effectiveness of other therapies and financial reimbursement. For example: if someone experienced enough benefit from treatment A, they will not seek for treatment B. While people that did not have enough pain relief with treatment A, will continue seeking for treatment B. The themes are further outlined in the following section.

**1. Generic aspects to seeking care.** Several factors are not specific to chiropractic care or LBP but might differ in older adults compared to their younger counterparts. These factors played a role for the participants during the decision-making process. For example, proximity to a chiropractic clinic and waiting time, were important factors. In addition, the LBP sufferers saw the ineffectiveness of other therapies as a major facilitator, meaning the lack of benefit from earlier treatment was a motivator to search elsewhere.

> "...at a certain point, everything seems to be fine and then I don't go looking for all kinds of others things."

(participant 10)

The focus group members agreed that the accessibility could be a barrier for older adults. Older adults are more likely to depend on someone else to bring them to the chiropractor, as chiropractors might not be located as close by as other therapists.

During the focus group, the issue was raised that older adults might not seek help for their back pain as quickly as their younger counterparts as they see it as a normal aging process or they have other conditions requiring priority of care.

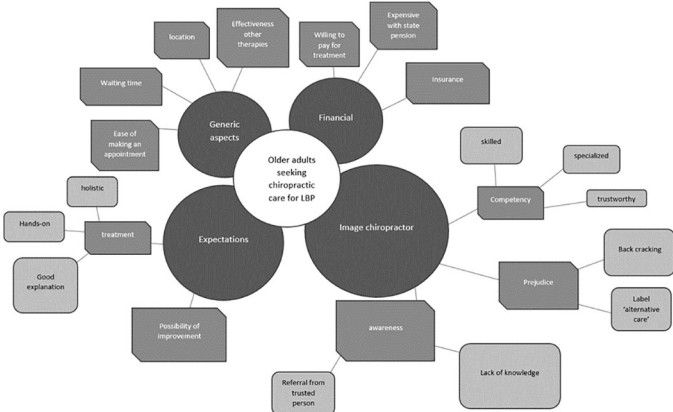

**Fig 2. The four identified themes of older adults seeking chiropractic care for their LBP.** The dark circles represent the themes, the hexagons represent the subthemes and the rounded rectangles represent the code families.

*"But because they have so many other health issues going on, low back pain can get a bit snowed under. And um, then other complaints become much more important. So, I think [. . .] people just learn to cope with it and move on."*

(Focus group, physiotherapist)

**2. Financial.** Multiple participants (participants 2,4 and 6) stated that the treatments were fairly expensive as they were dependent upon the state for their pension. Some said they would think twice or wouldn't have gone to see a chiropractor at all if their insurance had not reimbursed part of the costs, while others said they would have gone to see a chiropractor regardless of the reimbursement.

*[if it's not covered by insurance] "then I'd still consider doing it. But could I manage it?. . . because I'd have to use my savings. [. . .] So, I'd like to, but then I'd have to really figure it all out, to see if it's possible, because it can really end up being expensive."*

(participant 2)

The focus group members agreed that older adults have a more limited income than their working counterparts, which makes it harder for them to pay for the treatments out-of-pocket prior to reimbursement from the health insurance.

**3. Expectation.** The most common reported facilitator was the possibility of improvement. The participants hoped that their pain could be relieved by chiropractic care.

*"And then I thought, well, I've never actually tried that. And if there's something I've never tried, I should find out if it works, see if it's any different"*

(participant 2)

The participants expected a holistic approach from the chiropractor rather than focusing solely on the low back. They also expected a good explanation about their LBP and a hands-on treatment.

*"Well, that he really. . . I've never had manual treatment for my back."*

(participant 2)

One focus group member reported that some older adults prefer the holistic or alternative approach while another had the experience that her patients preferred hands-on treatments.

*"I've noticed that sometimes people come in who, um, yeah, have the feeling that chiropractors look at things in a really holistic way, and, well I gues they just have a preference for that."*

(Focus group, chiropractor)

**4. Image of the chiropractor.** The main barrier that was reported by the participants was the awareness. The participants which had not made an appointment with a chiropractor stated that they did not know what conditions a chiropractor treated or how. Most of the participants with an appointment stated that they only became aware of chiropractic care recently,

mostly through a trusted person. Most of these trusted persons were acquaintances of the participants, but also other healthcare professionals.

"Well, you wouldn't believe it, but I didn't know there was such a thing."

(participant 7)

Furthermore, there was some prejudice towards the chiropractor. Some participants mentioned they had heard of 'back-crackers' before. However, none of them had a negative association with the 'back-cracking' as they trusted the practitioner to be skilled and trained to do this safely, while others did not mention it. One patient who did not have an appointment with the chiropractor reported that she did not know what chiropractic care was, but that she expected the chiropractic treatment would not be suitable for her complaint.

"So, I assume [the chiropractor] would do me more harm than good. But I actually have no idea what a chiropractor could actually do about it really."

(participant 9)

The label 'alternative care' that chiropractic care carries within the Dutch healthcare system was thought to be a barrier. Some participants stated that this made them hesitate, others did not understand why chiropractic care was considered alternative care, while others were happy to choose an alternative care modality to the mainstream healthcare because they had not been helped sufficiently up to that moment.

*"How 'alternative' is this going tob e? Well, it turns out it's really not that bad, in my opinion. . . I also didn't expect (to find) that it was based on such a long period of study."*

(participant 8)

Most of the participants did some research on chiropractic care before making their appointment. The chiropractic education often impressed or surprised them. From information gathered on the websites of the local clinic and the national chiropractors association, they had a vision of chiropractors being competent: skilled, specialized, and trustworthy.

The focus group members agreed that awareness could be a barrier. They thought that patients as well as healthcare practitioners lacked a clear image of what a chiropractor does. They stated that older adults put more value on referrals or advice from healthcare practitioners. Therefore, they stated that the lack of knowledge among healthcare practitioners could be a barrier, as these practitioners will not refer to chiropractic care, a relatively unknown therapy to them. They also mentioned that healthcare practitioners do not refer because it is not recommended in the guidelines and they prefer referring to active treatments which are recommended by the guidelines.

"That's why I never actually refer patients to a chiropractor, because I have no earthly idea which patients should specifically go to a chiropractor."

(Focus group, physiotherapist)

It was also expected by the focus group members that older adults would also be hesitant to visit a chiropractor because of osteoporosis or other co-morbidities. Furthermore, the focus

group members stated that some of their patients consciously look for a therapy that is labelled alternative care.

## Discussion

We identified four themes regarding the barriers and facilitators for older adults to seek chiropractic care for their LBP (i.e. generic, financial, expectation, and image of the chiropractor). There were generic aspects, such as the location and the effectiveness of previous therapies, factors generally considered barriers and facilitators for care-seeking behaviour, not specific to chiropractic care. Financial factors impeded some older adults, while the expectation that the chiropractor could provide relief of their pain was a key facilitator. The image of the chiropractor both impeded and facilitated older adults seeking care from a chiropractor. That is, the prejudice against the terms 'alternative care' and 'back-cracking' were rarely identified as barriers because these terms were seen as outdated by most participants. The main barrier for older adults to seek care from a chiropractor was the lack of knowledge about chiropractic care. Older adults are often not aware of the existence of a chiropractor and other healthcare professionals often do not refer to a chiropractor because of their lack of knowledge on the care chiropractors give.

To our knowledge, this is the first study investigating the barriers and facilitators experienced by older adults for seeking chiropractic care for their LBP before their initial visit. Other qualitative studies examined the perspectives of older adults on LBP and the care they received from the chiropractor [2, 8]. One of the participants [8] stated they consider chiropractic care a primary treatment modality for LBP. However, the role of chiropractic care in the health care system can differ greatly between countries. Our study demonstrates lack of awareness of chiropractic care in the Netherlands. This would suggest that the results of previous studies, conducted in other countries and settings are not generalizable, and are context specific. A previous qualitative study [11] focused on barriers and facilitators for manual therapy referral for neck pain and identified similar barriers and facilitators as we identified, such as awareness, experience with other therapies, and financial aspects. However, the similarity may be somewhat surprising because our study differs in the age of the participants, therapist, and location of the pain, suggesting that these factors are not unique and may be generalizable to a broader group of patients seeking care from a therapist providing 'manual therapy'.

Perhaps the most important barrier to seeking was a general unfamiliarity regarding chiropractic care and specifically ignorance regarding manual therapies [11]. This lack of knowledge and experience has also been seen within the chiropractic setting previously [21, 22]. The members of the focus group agreed that healthcare providers are reluctant to refer to a chiropractor as they see chiropractic care as a passive form of therapy. Also, they lacked sufficient knowledge on which patients to refer. It can therefore be argued that health care provider exposure to chiropractic care may improve the patient care seeing pathway.

Patients with prior experience with chiropractic care were excluded because those with a prior (positive) bias are likely to view the barriers and facilitators to chiropractic care differently than those who are naïve to care. The hesitance of physiotherapists to refer to a chiropractor was also seen in previous studies [11], in which it was described as 'being vulnerable' and 'letting someone go'. It is also likely that referring is associated with a shrinking professional boundary and income [23]. Although healthy collaboration between various therapists can facilitate access to chiropractic care for older adults, active collaboration is difficult to broker and maintain and consequently is currently limited [24].

Our group had expected to observe prejudice regarding 'back-cracking' as a complication to manual care provided by chirorpactors and an issue that would emerge as a barrier to care

seeking. This was however not the case. Rather our participants, whilst being familiar with this prejudice, viewed this as an outdated image of the chiropractor.

## Strengths and challenges

All studies have strengths and limitations and our study is no exception. The study design included triangulation of perspectives and methods, which is a strength of this paper. We focussed on the perceived barriers and facilitators by older adults but also examined whether different professionals working closely with older adults agreed with the identified factors. Besides the interviews with the older adults, we also conducted a focus group with professionals and sent out member checks to the older adults after the interviews to verify the interpretation of the interviews to increase validity.

Patients with prior experience with chiropractic care were excluded because those with a prior (positive) experience are likely to view the barriers and facilitators to chiropractic care differently than those who are naïve to care.

The most important limitations include the following: Firstly, the interviews were conducted by an early-career researcher, still developing her interviewing skills; this, and the fact that she is a practicing chiropractor, could have resulted in an inherent bias during questioning. Secondly, the majority of the researchers of this paper are practicing chiropractors, and their perspectives could have influenced and driven the data analysis. Despite reaching data saturation, more information could be available if we would have interviewed more individuals due to the 'dynamic diversity' of the population.

The decision to conduct the interviews over video call was influenced by the COVID-19 pandemic [25, 26]. To protect the more fragile participants, we chose to conduct video calls rather than live interviews. This choice, however, potentially led to selection bias, because some participants were excluded if they did not have access to a computer. However, we noticed that even some of the digitally challenged managed to participate with the help of their family. Therefore, we are of the opinion that this bias was limited.

Our participants represented a spread of socio-demographic characterics such as age, gender, sociodemographic status, and functioning and therefore, our findings might be transferrable to other older adults in the Dutch setting. However, undoubtedly, groups of older adults might express differing opinions and views due to the 'dynamic diversity' [27]. When interpreting the results of this study, caution has to be taken to extrapolate the results.

## Conclusion

We examined what factors limit and facilitate older adults when seeking chiroprctic care. Older adults are often unaware of the possibility of chiropractic care for their LBP, however this factor might not be limited to this age group. The results of this explorative study can help us improve the accessibility to chiropractic care for older adults. However, how awareness of chiropractic care could best be implemented requires further investigation.

## Supporting information

**S1 Appendix. Topic guide (translated).**
(DOCX)

## Acknowledgments

We would like to thank *Chiropractie Roosenberg*, *Chiromotion*, *Chiropractie Borger*, *Chiropractie Zaltbommel* and *Chiropractie rugkliniek Heerlen* for their help recruiting participants.

## Author Contributions

**Conceptualization:** Lobke P. De la Ruelle, Annemarie de Zoete.

**Formal analysis:** Lobke P. De la Ruelle, Annemarie de Zoete.

**Funding acquisition:** Sidney M. Rubinstein.

**Methodology:** Lobke P. De la Ruelle, Cornelius Myburgh, Hella E. Brandt.

**Supervision:** Annemarie de Zoete, Cornelius Myburgh, Hella E. Brandt, Sidney M. Rubinstein.

**Writing – original draft:** Lobke P. De la Ruelle.

**Writing – review & editing:** Annemarie de Zoete, Cornelius Myburgh, Hella E. Brandt, Sidney M. Rubinstein.

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
