## [Decision Letter · Decision Letter 0]

14 Dec 2022

PONE-D-22-29577What are the perceived barriers and facilitators for chiropractic care in older adults with low back pain?PLOS ONE

Dear Dr. De la Ruelle,

Thank you for submitting your manuscript to PLOS ONE. After careful consideration, we feel that it has merit but does not fully meet PLOS ONE’s publication criteria as it currently stands. Therefore, we invite you to submit a revised version of the manuscript that addresses the points raised during the review process.

I am sending you the reviewer comments from a single reviewer, in order to expedite the process. I am confident in doing this due to the high level of detail and attention in the reviewers’ comments.

Please address all points, and then I look forward to seeing your revised manuscript.

We look forward to receiving your revised manuscript.

Kind regards,

Tim Alex Lindskou

Academic Editor

PLOS ONE

Journal Requirements:

"This study is funded by European Centre for Chiropractic Research Excellence (ECCRE), located in Odense, Denmark (grant number 01-20160NL/MvT) as well as funded by the Nederlandse Chiropractoren Associatie (NCA), located in Emmeloord. "

4. Thank you for stating the following in the Funding Section of your manuscript: 

"This study is funded by European Centre for Chiropractic Research Excellence (ECCRE), located in Odense, Denmark (grant number 01-20160NL/MvT) as well as funded by the Nederlandse Chiropractoren Associatie (NCA), located in Emmeloord. "

"This study is funded by European Centre for Chiropractic Research Excellence (ECCRE), located in Odense, Denmark (grant number 01-20160NL/MvT) as well as funded by the Nederlandse Chiropractoren Associatie (NCA), located in Emmeloord. "

"Four authors (LDLR, ADZ, CM and SMR) are chiropractors who work in clinical practice, but have no direct financial interests linked with this survey."

7. Please upload a new copy of Figure 2 as the detail is not clear. Please follow the link for more information: https://blogs.plos.org/plos/2019/06/looking-good-tips-for-creating-your-plos-figures-graphics/" https://blogs.plos.org/plos/2019/06/looking-good-tips-for-creating-your-plos-figures-graphics/

Reviewers' comments:

Reviewer's Responses to Questions

**Comments to the Author**

1. Is the manuscript technically sound, and do the data support the conclusions?

Reviewer #1: Partly

2. Has the statistical analysis been performed appropriately and rigorously? 

Reviewer #1: I Don't Know

3. Have the authors made all data underlying the findings in their manuscript fully available?

Reviewer #1: No

4. Is the manuscript presented in an intelligible fashion and written in standard English?

Reviewer #1: No

5. Review Comments to the Author

Reviewer #1: This study is a good early work by a young investigator, exploring the perspectives of Dutch older adults about seeking chiropractic care. The background would be strengthened by inclusion of more detail about role of chiropractic in Dutch healthcare system - funding, number of chiropractors, training - this would serve to educate readers about the profession in this context - the lack of which info was mentioned as a barrier by patients and providers alike. Other comments to improve the paper are as follows. There are many, but please, GO FOR IT! You can make these revisions, the paper will be clearer, and you'll make a great contribution to the literature - and be set for your next study! You can do this!!

1) Lines 90-95 - References 2-4 seem to have dropped off, making credibility of statements difficult to ascertain.

2) Lines 103-104 - Qualitative study healthcare seeking back pain older adults - There's an extensive qualitative literature on this topic, enough that there's a meta-synthesis and systematic review on care seeking for LBP. You might not find it by looking for 'barriers and facilitators' alone, but it's there - AND there's info about this in regards to chiropractic care, often in the context of interprofessional care (MD/DC or DC/PT) - as well as disabling pain (Makris et al). This reviewer is very familiar with this work, and encourages the team to look more closely at this literature to inform both the background and the discussion of this paper (Essential revision).

3) Lines 134-137 - Design. No references are provided for the numerous methodologies mentioned: exploratory qualitative, case study, naturalistic design, constructivist lens, factor(analysis?), and lived experience (typically used to describe phenomenology - which is not in alignment with a constructivist lens). Pick a design (exploratory/descriptive qualitative is sufficient), include references that guided this approach, and include sufficient detail that another qualitative researcher can tell what was done and how this approach informed the thinking/data analysis. (Essential revision) Move COREQ statement up here, too, to allow reader to know how presentation is being structured.

4) Lines 138-141 - Theoretical approach - thematic analysis in an analytic approach than a theoretical lens. You might have a theoretical approach listed above from a disciplinary lens, which will require more explication and reference here. What would be important to include is some theoretical lens through which you viewed old age - disengagement, continuity, activity theories? Integrity vs despair? Others? HOW the research team view old people and the aging process will deeply influence how questions were asked, and the themes discovered- for example, 55 year old people are considered old rather than midlife for inclusion purposes...this is a lens through which aging is viewed. This is an essential revision that must be addressed.

5) Lines 145 - Include any pertinent references to the BACE-C study, as well as some key eligibility criteria to larger study.

6) Lines 146-149 - Consider how these exclusions might have biased your findings in the discussion. For example, the decision to exclude patients who have received previous chiropractic care does not UNBIAS this study - it just differently biases the study, because people who had NOT received chiropractic care for at least 55 years have STRONG OPINIONS about chiropractic care - it kept them off the table! This decision is a limitation, because we now don't know what is a barrier or facilitator for people who are positively inclined to use chiropractic - so cannot build on what works!

7) Lines 159 - Tech reference for Zoom company (as for any scientific/tech company) and rationale as to why this technology was used over other similar technologies.

8) Line 176 - How were interviews transcribed - by the Zoom software, or by the investigators? Provide reference to support this decision. How was translation from Dutch to English handled for publication? This should be in methods not results and include the process and who completed this task.

9) Data collection - no reference are provided for decisions make in stage 1 or 2. Methods need references.

10) Interviews - how was oversight provided for this first time interviewer? Which team member handled this training and oversight?

11) Line 180 - Purposive sampling needs reference. And more description as to how this happened.

12) Line 199 - MaxQDA needs company details. How was intercoding agreement assessed?

13) Line 201 - Member check needs reference.

14) Data analysis - more detail on thematic analysis are needed as it's unclear who did what and how codes/themes were generated (particularly given the many theoretical/methodological frames mentioned).

15) Data analysis - there's no mention about how participants were managed as "cases" to understand individual patterns before group patterns - this means this study isn't a case study. So that's one design it will be easy to let go of in the revision!

16) Figure 1 - Need more information to interpret this figure, including details that this is about care seeking! It's unclear why words are in different sizes, what the various arrows mean. Specific discussion about why chiropractors did not engage in any referral would be helpful. The figure is also messy from a 'time' perspective - it's unclear if this is 1 patient who did all this (a case) or a general sum up, but it looks like GPs are late in the game (reading figure for 'time' from left to right), but they probably come first. Make more meaning with figure and describe better in text. Figure legends are also needed. Lines 248 - this info about how patients found chiropractor could be included in figure to better explicate healthcare seeking (walked past clinic, flyer, etc).

17) Table 1 - Inclusion of detail on which providers had received care from would offer additional info to make Figure 1 and 2 clearer.

18) Figure 2 - This figure is also very messy and almost unreadable - formatting per journal instructions are needed. The 4 primary themes/factors are floating in space - link these 4 to a central theme, which is 'care seeking behavior for LBP'. From there, there's no legend that tells the reader how to read the figure. What's a circle, hexagon, or rectangle mean? Are there NO links between the 4 main themes? What's a barrier? What's a facilitator? Make figure clear on all levels. Are some themes from patients and other from focus groups with providers? What's the distinction? Are some from the chiropractic patients - or those that did not choose the chiropractor?

19) Reconsider "generic" as a theme and consider "patient-centered" or "patient factors". These were about patients' lives and should be honored as such.

20) Discussion - more references needed throughout.

21) Availability of data statement is not same in manuscript as in cover.

22) Did the lead author not take part in conceputalization of study?

23) Requires copyediting for English language; non-standard referencing for journal.

6. PLOS authors have the option to publish the peer review history of their article (what does this mean?). If published, this will include your full peer review and any attached files.

Reviewer #1: **Yes: **Stacie A. Salsbury, PhD, RN

---

## [Author Response · Author response to Decision Letter 0]

9 Feb 2023

Amsterdam, The Netherlands

February 7th, 2023

Manuscript: CHMT-D-22-00008

What are the perceived barriers and facilitators for chiropractic care in older adults with low back pain?

Dear Editor-in-Chief,

Thank you for the opportunity to reconsider publication of our manuscript following revision and modification.

Below you will find our response to the comments by the editor and reviewer. The necessary changes have been made to the manuscript and have been uploaded in the system. These changes are clearly indicated in the editorial mode (‘track changes’). 

The ‘Role of Funder’ statement is the following: 

This study is funded by European Centre for Chiropractic Research Excellence (ECCRE), located in Odense, Denmark (grant number 01-20160NL/MvT) as well as funded by the Nederlandse Chiropractoren Associatie (NCA), located in Emmeloord. The funders had no role in study design, data collection and analysis, decision to publish, or preparation of the manuscript.

The ‘competing interests’ statement is the following: 

Four authors (LDLR, ADZ, CM and SMR) are chiropractors who work in clinical practice, but have no direct financial interests linked with this survey. This does not alter our adherence to PLOS ONE policies on sharing data and materials.

The ‘data availability’ statement is the following: 

All relevant data are within the manuscript and its supporting information files. The transcripts of the interviews and focus group cannot be shared publicly due to privacy reasons as participants can be identified. This is in accordance with European Data Protection Regulation (GDPR) (https://libguides.vu.nl/rdm/gdpr-privacy). Individual participants’ data are available upon request from the principal investigator (Sidney Rubinstein, s.m.rubinstein@vu.nl) for researchers who meet the criteria for access to confidential data). 

We look forward to your reaction. On behalf of all authors sincerely,

Lobke De la Ruelle 

Correspondence:

Lobke De la Ruelle 

Department Health Sciences, 

Faculty of Science,

Vrije Universiteit Amsterdam, 

Van der Boechorststraat 7

1081 BT Amsterdam, 

The Netherlands

l.p.delaruelle@vu.nl

---

## [Decision Letter · Decision Letter 1]

14 Mar 2023

The Perceived Barriers And Facilitators For Chiropractic Care In Older Adults With Low Back Pain; Insights From A Qualitative Exploration In A Dutch Context

PONE-D-22-29577R1

Dear Dr. De la Ruelle,

We’re pleased to inform you that your manuscript has been judged scientifically suitable for publication and will be formally accepted for publication once it meets all outstanding technical requirements.

Kind regards,

Tim Alex Lindskou

Academic Editor

PLOS ONE

Additional Editor Comments (optional):

Reviewers' comments:

Reviewer's Responses to Questions

**Comments to the Author**

1. If the authors have adequately addressed your comments raised in a previous round of review and you feel that this manuscript is now acceptable for publication, you may indicate that here to bypass the “Comments to the Author” section, enter your conflict of interest statement in the “Confidential to Editor” section, and submit your "Accept" recommendation.

Reviewer #1: All comments have been addressed

2. Is the manuscript technically sound, and do the data support the conclusions?

Reviewer #1: Yes

3. Has the statistical analysis been performed appropriately and rigorously? 

Reviewer #1: Yes

4. Have the authors made all data underlying the findings in their manuscript fully available?

Reviewer #1: No

5. Is the manuscript presented in an intelligible fashion and written in standard English?

Reviewer #1: Yes

6. Review Comments to the Author

Reviewer #1: Very good revisions. All concerns addressed. Minor typos throughout which should be caught during galleys. Good luck!

7. PLOS authors have the option to publish the peer review history of their article (what does this mean?). If published, this will include your full peer review and any attached files.

Reviewer #1: **Yes: **Stacie A Salsbury

---

## [Editor Report · Acceptance letter]

30 Mar 2023

PONE-D-22-29577R1 

The Perceived Barriers And Facilitators For Chiropractic Care In Older Adults With Low Back Pain; Insights From A Qualitative Exploration In A Dutch Context 

Dear Dr. De la Ruelle:

I'm pleased to inform you that your manuscript has been deemed suitable for publication in PLOS ONE. Congratulations! Your manuscript is now with our production department. 

Kind regards, 

on behalf of

Dr. Tim Alex Lindskou 

Academic Editor

PLOS ONE